# DNA FISH Diagnostic Assay on Cytological Samples of Thyroid Follicular Neoplasms [note 1]

**DOI:** 10.3390/cancers12092529

**Published:** 2020-09-06

**Authors:** Philippe Vielh, Zsofia Balogh, Voichita Suciu, Catherine Richon, Bastien Job, Guillaume Meurice, Alexander Valent, Ludovic Lacroix, Virginie Marty, Nelly Motte, Philippe Dessen, Bernard Caillou, Abir Al Ghuzlan, Jean-Michel Bidart, Vladimir Lazar, Paul Hofman, Jean-Yves Scoazec, Adel K. El-Naggar, Martin Schlumberger

**Affiliations:** 1Department of Medical Biology and Pathology, Gustave Roussy, Université Paris-Saclay and Experimental and Translational Pathology Platform, CNRS UMS3655-INSERM US23 AMMICA, 94805 Villejuif, France; zsofia.balogh@gustaveroussy.fr (Z.B.); voichita.suciu@gustaveroussy.fr (V.S.); catherine.richon@gustaveroussy.fr (C.R.); bastien.job@gustaveroussy.fr (B.J.); guillaume.meurice@gustaveroussy.fr (G.M.); alexander.valent@gustaveroussy.fr (A.V.); ludovic.lacroix@gustaveroussy.fr (L.L.); virginie.marty@gustaveroussy.fr (V.M.); nelly.motte@gustaveroussy.fr (N.M.); philippe.dessen@gustaveroussy.fr (P.D.); bernard.caillou@gustaveroussy.fr (B.C.); abir.alghuzlan@gustaveroussy.fr (A.A.G.); jean-michel.bidart@gustaveroussy.fr (J.-M.B.); vladimir.lazar@gustaveroussy.fr (V.L.); jean-yves.scoazec@gustaveroussy.fr (J.-Y.S.); 2Laboratory of Clinical and Experimental Pathology and Biobank, Pasteur Hospital, 06002 Nice, France; hofman.p@chu-nice.fr; 3Department of Pathology, The University of Texas MD Anderson Cancer Center, Houston, TX 77030, USA; anaggar@mdanderson.org; 4Department of Endocrinology, Gustave Roussy, Université Paris-Saclay, 94805 Villejuif, France; martin.schlumberger@gustaveroussy.fr

**Keywords:** thyroid, fine-needle aspiration, indeterminate cytology, follicular neoplasms, array comparative genomic hybridization, fluorescent in situ hybridization

## Abstract

**Simple Summary:**

Cytopathology cannot distinguish benign from malignant follicular lesions in 20–30% of cases. These indeterminate cases includes the so-called follicular neoplasms (FNs) according to The Bethesda System for Reporting Thyroid Cytopathology. Frozen samples from 66 classic follicular adenomas (cFAs) and carcinomas (cFTCs) studied by array-comparative genomic hybridization identified three specific alterations of cFTCs (losses of 1p36.33-35.1 and 22q13.2-13.31, and gain of whole chromosome X) confirmed by fluorescent in situ hybridization (FISH) in a second independent series of 60 touch preparations from frozen samples of cFAs and cFTCs. In a third independent set of 27 cases of already stained pre-operative fine-needle aspiration cytology samples diagnosed as FNs and histologically verified, FISH analysis using these three markers identified half of cFTCs. Specificity of our assay for identifying cFTCs is higher than 98% which might be comparable with *BRAF*^600E^ testing in cases of suspicion of classic papillary thyroid carcinomas.

**Abstract:**

Although fine-needle aspiration cytology (FNAC) is helpful in determining whether thyroid nodules are benign or malignant, this distinction remains a cytological challenge in follicular neoplasms. Identification of genomic alterations in cytological specimens with direct and routine techniques would therefore have great clinical value. A series of 153 cases consisting of 72 and 81 histopathologically confirmed classic follicular adenomas (cFAs) and classic follicular thyroid carcinomas (cFTCs), respectively, was studied by means of different molecular techniques in three different cohorts of patients (pts). In the first cohort (training set) of 66 pts, three specific alterations characterized by array comparative genomic hybridization (aCGH) were exclusively found in half of cFTCs. These structural abnormalities corresponded to losses of 1p36.33-35.1 and 22q13.2-13.31, and gain of whole chromosome X. The second independent cohort (validation set) of 60 pts confirmed these data on touch preparations of frozen follicular neoplasms by triple DNA fluorescent in situ hybridization using selected commercially available probes. The third cohort, consisting of 27 archived cytological samples from an equal number of pts that had been obtained for preoperative FNAC and morphologically classified as and histologically verified to be follicular neoplasms, confirmed our previous findings and showed the feasibility of the DNA FISH (DNA fluorescent in situ hybridization) assay. All together, these data suggest that our triple DNA FISH diagnostic assay may detect 50% of cFTCs with a specificity higher than 98% and be useful as a low-cost adjunct to cytomorphology to help further classify follicular neoplasms on already routinely stained cytological specimens.

## 1. Introduction

Thyroid nodules are a frequent finding, as they are palpable in approximately 5–10% of adults and detectable in 50–70% of people over the age of 60 by means of ultrasonography (US) [1,2,3]. However, malignant nodules are rare (5–10%), and the prognosis of patients with well-differentiated carcinomas is usually equally excellent for both sexes [4,5,6,7,8,9]. The triage of patients who are the best candidates for an initial surgical treatment based on evidence-based strategies is therefore needed to accurately identify preoperatively malignant thyroid nodules [1,10].

The most reliable initial diagnostic tool for the evaluation of thyroid nodules is the cytomorphologic study of materials obtained by fine-needle aspiration cytology (FNAC), preferentially guided by US [1] and reported according to The Bethesda System for Reporting Thyroid Cytopathology (TBSRTC) [11,12].
In 70–80% of all cases, FNAC provides a definitive diagnosis of benign lesions or papillary thyroid cancer (PTC) [1]. However, a meta-analysis published in 2012 reported that cytopathology alone cannot reliably distinguish benign from malignant follicular lesions in 20–30% of cases [13]. Such cases correspond to one of the three following TBSRTC indeterminate categories, the so-called “gray zone”: atypia or follicular lesion of undetermined significance (AUS/FLUS: category III), follicular or suspicious for a follicular neoplasm (FN/SFN: category IV) and suspicious for malignancy (SM: category V), the latter being mainly suspicious for PTC [11,12]. In addition to the difficulty in segregating follicular neoplasms into classic follicular adenoma (cFA) and malignant lesions, such as classic follicular thyroid carcinoma (cFTC), encapsulated follicular variant of papillary carcinoma (eFVPTC) and infiltrative follicular variant of papillary carcinoma (iFVPTC) [11], recognizing the recent entity known as noninvasive follicular tumor with papillary-like nuclear features (NIFTP) is even more challenging [12,14,15]. This is largely due to the current reliance on the established architectural criteria for histopathological examination of the surgical specimen [16].

Since FNAC has limitations in providing a definitive diagnosis for some nodules, management of the corresponding patients is hampered. For example, surgery may be unnecessary in cases of benign disease or suboptimal for patients with thyroid cancer who will need a second completion surgery. Consequently, identification of additional criteria to refine the use of FNAC for the diagnosis of thyroid malignancy has received considerable attention in recent years as a part of efforts to optimize medical and surgical management of patients, to minimize overtreatment and to reduce unnecessary surgeries along with their subsequent risks and health care burden [17,18,19].

In 2014, The Cancer Genome Atlas (TCGA) research network identified and reported recurrent gene mutations and fusions in PTC [20]. Tumors with true papillary structure were dominated by *BRAF*^V600E^ mutations and *RET* kinase fusions and were designated as being *BRAF*^V600E^-like, whereas iFVPTCs were conversely dominated by *RAS* mutations and were designated as being *RAS*-like [21]. This was later confirmed by an independent study [22]. However, few somatic molecular abnormalities have been shown to be diagnostically useful [23,24]. The best example is the *BRAF*^V600E^ mutation observed in approximately half of classic PTCs (cPTCs), in a higher percentage of the tall cell and hobnail PTC variants, and in FVPTCs. However, *BRAF*^V600E^ mutation is virtually not present in cFTCs [25]. Similarly, gene fusions between *RET* and different *PTC* partners in PTCs, including the *RET/PTC3* rearrangement which is more common in radiation-induced thyroid cancer [26], are not observed in cFTCs. In addition, the large spectrum of follicular-pattern benign and malignant tumors that share common genetic alterations is striking. For example, cFA, cFTC, NIFTP, eFVPTC, iFVPTC, cPTC and Hürthle cell carcinoma (HCC) may carry *RAS* point mutations, and all except for eFVPTC may carry *DICER1* alterations [25]; the *PPARγ*-*PAX8* gene fusion may be observed in cFA, cFTC and FVPTC as well as the *THADA* gene fusion in cFA, cFTC and NIFTP [25].

Known driver mutations in cFTCs correspond mainly to: (i) point mutations of the three human *RAS* genes (*NRAS*, *HRAS* and *KRAS* by frequency order) at codons 61 and, less frequently, at codons 12 or 13 in close to half of all cases. Other gene mutations may occur in *PTEN*, *DICER1*, *EIF1AX* (10–15%, each), *PIK3CA* (5–10%), *IDH1* and *IDH2* (<5% each), with *BRAF*^K601^ mutations being rare; (ii) large-scale alterations also occur, such as *PAX8-PPARγ* (10–20%) and *THADA-IGF2BP3* (<5%) gene fusions [27]. Here, again, most genetic alterations, including *PIK3CA* and *PTEN* mutations and *PPARγ* fusions with other partners, such as *CREB3L2,* may also be observed in some cases of cFA, FVPTC, NIFTP and Hürthle (oncocytic) cell neoplasms [27], precluding their use in a diagnostic test [28].

Although traditionally considered variants of follicular neoplasms, Hürthle cell neoplasms now represent a different entity according to the 2017 WHO classification of tumors of endocrine organs [16], with a distinct pathogenesis and clinical behavior [29,30,31,32]. They were not included in our study, even though cytomorphology is usually able to detect these tumors [33,34,35,36]. Similarly to the differential diagnosis between cFA and cFTC, demonstration of capsular and/or vascular invasion is required to ascertain malignancy [37,38,39,40,41].

Our hypothesis was that an in-depth analysis of a large series of cFA and cFTC genetic landscapes could potentially help to identify a combination of large-scale alterations, such as somatic copy number variations (sCNVs) characteristic of cFTC, and that these novel and validated genetic biomarkers would be critical for developing simple and direct tests that are applicable to cytological specimens to complement cytomorphology. Thus, we first studied two independent (training and validation) sets of histologically confirmed samples from two comprehensive cancer centers (Gustave Roussy (GR), Villejuif, France, and the University of Texas MD Anderson Cancer Center (UT MDACC), Houston, TX, USA): these two sets were frozen tissues from 126 pts with cFA (*n* = 59) and cFTC (*n* = 67). A third set comprising 27 stained FNAC smears (27 pts), morphologically preoperatively classified as FN/SFN according to TBSRTC and then histologically verified on the corresponding postoperative surgical specimens, represented the feasibility set.

## 2. Materials and Methods

### 2.1. Study Design, Patients and Tissue Sample Characterization

Figure 1 shows the flow chart of the study. All patients provided oral informed consent for the use of their samples. The institutional ethics and review board committees of GR and UT MDACC (respectively, #RT0106 and #DR11-0348 and MTA #2011-8521, approval date: 1 May 2020) approved this study. All frozen histological sections and touch preparations were verified by senior pathologists (Bernard Caillou and Philippe Vielh), and these materials showed that the tumor cell content was at least 80%. The initial diagnosis of cFTC performed on paraffin-embedded histopathological samples according to the 2004 WHO classification [37] was kept (invasion of the capsule only representing minimally invasive cFTC as opposed to widely invasive cFTC). These initial diagnoses were not retrospectively reclassified and stratified according to the most recent 2017 WHO classification of tumors of endocrine organs [16] and the number of veins (<4 vs. ≥4) invaded [38].

As shown in Figure 1, the study comprised three cohorts representing the training, validation and feasibility sets. The first cohort (training set) to be analyzed by aCGH consisted of 150 snap-frozen tumor and matched normal tissue samples, whenever possible (see also the first paragraph from Section 2.4.) from 87 pts. After exclusion of 21 cases (Figure 1), the training set consisted of 66 cases of follicular neoplasms corresponding to 33 cFAs and 33 cFTCs (Table 1). Of these 66 cases, 44 with follicular neoplasms underwent further DNA FISH analysis of touch preparations from frozen tissues (Figure 1). The second cohort (validation set) consisted of 68 samples (68 pts) corresponding to 26 cFAs and 34 cFTCs (Table 1) from 60 pts studied by DNA FISH on touch preparations from the corresponding frozen tissues. The third cohort (feasibility set) consisted of a series of 27 archived preoperative thyroid FNAC specimens (Figure 1), reported as FN/SFN by cytology according to TBSRTC [11]. Histopathological examination of the corresponding postresection surgical specimens, paired with those of the training and validation sets, showed cFTC and cFA in 14 and 13 cases, respectively. Cases with at least two cytology slides available (to maintain one in the archives for medico-legal purposes) were retrieved, and one was further used for DNA FISH.

Table 1 summarizes the clinicopathological characteristics of the 126 follicular neoplasms studied in the training and validation sets. DNA ploidy analysis was performed whenever possible, as well as mutational analysis of the *RAS* family of genes, *CTNNB1* and *PIK3CA* genes by Sanger direct sequencing, and translocation analysis (*PAX8/PPARγ* and *RET/PTC1*) by reverse-transcriptase-polymerase chain reaction (RT-PCR).

### 2.2. DNA Ploidy Analysis

Frozen tumor tissue samples were quickly thawed and dissociated mechanically, and the DNA was stained with the BD CycleTEST PLUS DNA Reagent Kit (Beckton Dickinson, San Jose, CA, USA). DNA flow cytometry was performed on a Beckman Coulter Epics XL MCL flow cytometer equipped with a 488-nm argon laser (Beckman Coulter, Villepinte, France). The DNA diploid peak was located on DNA histograms according to an external standardization procedure using normal human lymphocytes. When a slight DNA abnormality was suspected, thawed normal human lymphocytes were added to the cell suspensions before DNA staining. The resulting DNA histograms were interpreted using MultiCycle AV software (Phoenix Flow Systems, San Diego, CA, USA). DNA indices (DIs) were computed as previously described [42]. DIs ranging from 0.9 to 1.1 corresponded to DNA diploidy; DIs between 0.6 and 0.9 and between 1.1 and 1.8 were classified as DNA aneuploid; DIs between 0.5 and 0.6 and between 1.8 and 2.2 were considered DNA haploid and DNA tetraploid, respectively. DNA multiploidy corresponded to cases with the presence of two or more DIs.

### 2.3. Nucleic Acid Extraction

DNA was extracted from paired frozen normal and tumor samples using the DNeasy^®^ Tissue Kit (Qiagen, Hilden, Germany) according to the manufacturer’s recommendations.

RNA extraction was performed using the TRIzol^®^ Reagent Protocol (Invitrogen, Carlsbad, CA, USA). Total RNA was quantified, and the purity was assessed using a Nanodrop ND-1000 spectrometer and a Bioanalyser-2100 (Agilent, Santa Clara, CA, USA).

### 2.4. Oligonucleotide aCGH Assay, Data Processing and Analysis

In 53 (80%) out of 66 cases (training set), the matched normal tissue sample was used as a reference for comparison with the corresponding tumor sample. For the remaining 13 (20%) cases in which no normal tissue was available, commercially available pooled sex-matched Human Genomic DNA (Promega, Madison, WI, USA) was used as the reference sample (see also Section 2.1 about study design in Materials and Methods).

Equal amounts (500 ng) of DNA from the tumor and normal samples were treated with the AluI and RsaI restriction enzymes (Fermentas, Euromedex, France) and labeled with cyanine (Cy)3-deoxyuridine triphosphate (dUTP) or Cy5-dUTP. Hybridization was carried out using Agilent 244K Whole Human Genome (G4411B) arrays (Agilent Technologies, Santa Clara, CA, USA) for 40 h at 65 °C in a rotating oven (Robbins Scientific, Mountain View, CA, USA) at 20 rpm. Hybridization was followed by the appropriate washing steps. Scanning was performed with an Agilent G2505C DNA Microarray scanner at 100% PMT with 5 µm resolution at 20 °C in a low ozone concentration environment. Quantitation of the Cy5 and Cy3 signals from the scans was performed with Feature Extraction v10.1 (Agilent) using the default parameters.

Raw aCGH signals and log2 (ratio) profiles were normalized and centered using an as-yet unpublished in-house method according to the dye composition (Cy5/Cy3) and local GC% composition. These profiles were segmented with the circular binary segmentation algorithm [43] through its implementation in the DNAcopy v1.30 package for R v2.15.1 using default parameters. DNA copy number imbalances were detected by considering a minimum of three consecutive probes and a minimal absolute amplitude threshold that was specific for each profile according to its internal noise. This specific internal noise was computed as one-fourth of the median of the absolute log2 (ratio) distances across consecutive probes on the genome. All aCGH coordinates in this study were mapped against the human genome as defined by the UCSC build hg19 (http://genome-euro.ucsc.edu/). Hierarchical clustering of samples was performed on segmented profiles using Pearson or Euclidean distance for whole-genome profiles or for selected genomic regions, respectively. Ward’s method was used in both cases. Differential analysis was performed using Wilcoxon’s sum-rank test and minimum differentiating were selected using an iterative approach: significant regions were filtered to keep regions for which all cFA samples were normal, then ranked by their increasing raw *p*-value; the significant region with the highest coverage of cFA was selected, then the next region covering the most new cFA samples not included in the first one, and so on. Frequency plots, clustering and heatmaps were all generated using R with in-house scripts. All statistical computations were performed using R with default packages. Genomic region annotation was performed by an in-house script using annotation resources available at the UCSC Tables repository (https://genome.ucsc.edu/cgi-bin/hgTables). Data are available under the Array Express accession number E-MTAB-1575.

### 2.5. Triple DNA FISH Assay

DNA FISH was performed on touch preparations of frozen tissues from 44 samples (training set, 44 pts) and 60 samples (validation set, 60 pts) and in a series of 27 (feasibility set, 27 pts) archived FNAC samples (Figure 1) with three commercially available directly labeled fluorescent probes (Vysis CEP X/Y, LSI TUPLE 1/LSI ARSA, Abbott Laboratories, Abbott Park, IL, USA; ZytoLight 1p36/1q25, ZytoVision, Bremerhaven, Germany): we named this combination the “Triple DNA FISH assay”.

On touch preparations, cells were fixed directly with a 9:1 mixture of methanol and acetic acid for 15 min and air-dried overnight. After enzymatic pepsin treatment, nuclear DNA was denatured in 70% formamide/2× SSC and dehydrated. The probes were heated for 5 min at 85 °C and hybridized overnight at 40 °C. Slides were washed with stringent buffer, air-dried and counterstained with DAPI.

On archived FNAC samples already stained using the May-Grünwald-Giemsa or Diff Quik methods, a slightly modified protocol was used. Briefly, coverslips were first removed using xylene, and cells were exposed to pepsin and dehydrated. Then, heat-denaturation was performed for 5 min at 80 °C. The glass slide was splitted in three different fields using small round coverslips and hybridization was carried out overnight at 40 °C. These archived FNAC samples were initially used for preoperative diagnostic purposes and then stored for a period of time ranging from 1 month up to 12 years. They corresponded to samples classified as FN/SFN according to TBSRTC (Bethesda category IV) [11,12] and to cFA or cFTC on the matching histological specimens.

The triple DNA FISH assay was interpreted by senior cytogeneticists (Alexander Valent, Zsofia Balogh) blinded to histopathological results. Preparations were observed with an epifluorescence microscope (Leica Biosystems, Nanterre, France), and images were captured with the CytoVision imaging station (Leica Biosystems, France). Only nuclei with unambiguous probe hybridization were scored to determine the number of signals per nucleus. At least 100 nuclei were analyzed for each hybridization.

### 2.6. Gene Mutation Analyses

Exons 2 and 3 of *KRAS* (NM_033360.2), *HRAS* (NM_005343.2) and *NRAS* (NM_005343.2), exons 11 and 15 of *BRAF* (NM_004333.4), exon 3 of *CTNNB1* (NM_001098209.1) and exons 1 and 21 of *PIK3CA* (NM_006218.2) were analyzed by direct Sanger sequencing after specific amplification by PCR, as previously described [44]. Sense and antisense sequences were screened for exonic alterations using SeqScape v2.5 software (Applied Biosystems, Foster City, CA, USA). All detected mutations were confirmed by at least one independent PCR.

### 2.7. Translocation Analyses

Total RNA (500 ng) was reverse-transcribed using Moloney murine leukemia virus reverse transcriptase in the presence of random primers (Applied Biosystems). Amplification of the RPLP0 housekeeping gene by quantitative real-time PCR was used as a control for the efficiency of the reverse transcription reaction. For each experiment, cDNA from samples known to be positive or negative for translocations was used as a control. *PAX8-PPARγ1* and *RET-PTC1* translocation detection assays were performed as previously described [45,46].

## 3. Results

### 3.1. Clinical, Histopathological and Biological Data

The clinicopathological and biological characteristics of the 126 pts included in the training (66 pts) and validation (60 pts) sets are summarized in Table 1. Histopathological subtypes [16] of cFA (*n* = 59) showed a microfollicular, macrofollicular and mixed architectural pattern of growth in 19, 13 and 27 pts, respectively (without fetal/solid type), whereas cFTCs (*n* = 67) with various growth patterns were minimally invasive (capsular invasion only) and widely invasive in 24 and 33 pts, respectively.

DNA flow cytometry was performed in the training and validation sets. As a whole, 37 (63%) out of 59 cFAs and 30 (58%) out of 52 cFTCs (15 of which were unsatisfactory for interpretation) were DNA diploid tumors. The other cFAs were DNA aneuploid, tetraploid and multiploid tumors in 10 (17%), four (7%) and eight (13%) cases, respectively (none were DNA haploid). The remaining cFTCs were DNA aneuploid, tetraploid and multiploid in 10 (19%), four (7%) and five (10%) cases, respectively, while three (6%) cases were DNA haploid.

There were no statistically significant differences (*p* < 0.05) in the clinical or biological data between the training with validation set. The only exception was that the 17q gain was more frequent in men and in cases with mutated genes of the *RAS* family (*p* < 0.05). One *BRAF*-mutated case was found in the training set.

### 3.2. Training Set on Frozen Tissue

By means of aCGH, sCNV was determined in a total of tumors from 66 pts (training set). Copy number gains were generally more frequent than losses, regardless of the tumor histological subtype, namely, minimally invasive (capsular invasion only) or widely invasive FTC. Figure 2A shows the global heatmap of the 66 clustered samples using the Pearson distance and Ward’s construction method. Three subgroups were observed (Figure 2A): one with complex abnormalities found in 17 (26%) out of 66 cases (13 (39%) out of 33 cFTCs and four (12%) out of 33 cFAs; one with a single loss of 22q found in a subpopulation of cFTCs (six (18%) out of 33 cases) and no cFAs; and one with no recurrent structural genetic anomalies, comprising 29 (88%) out of 33 cFAs and 14 (42%) out of 33 cFTCs. Figure 2B shows that the gains and losses for each chromosomal region were higher in cFTC than in cFA. Alterations found in both benign and malignant samples were gains of chromosomes 5, 7, 9, 12, 14, 16, 17 and 20 and losses of chromosomes 3, 4, 8, 11 and 21 (Figure 2B). No significant differences in terms of number or type of genomic anomalies were observed between minimally and widely invasive cFTCs. The heatmaps of the three discriminating chromosome arms/chromosomes (chromosomes 1p, 22q and X) displayed two clusters (Figure 3). The first cluster contained 18 (55%) out of 33 cFTCs and one (3%) out of 33 cFAs, and consisted of deletions of chromosome 1p or 22q and/or a gain of chromosome X. The second cluster included 32 (97%) out of 33 cFAs and 15 (45%) out of 33 cFTCs and exhibited no recurrent structural genetic abnormality or a gain of chromosome 22 as a distinguishing finding. Further analysis revealed that the three distinctive minimal common regions differentiating cFTCs from cFAs (Figure 4) were loss of 1p36.33-1p35.1 (five cases, 15%); loss of 22q13.2-22q13.31 (eight cases, 24%); and gain of chromosome X (eight cases, 24%). In the training set, we found 21 genetic alterations in 18 pts (one case had two aberrations: 1p loss plus 22q loss; one case had all three aberrations: 1p and 22q losses and X gain) (Figure 3). These abnormalities were exclusive to cFTCs in 55% of cases. Interestingly, X gains were always whole-chromosome gains (Figure 2A). The content of these three distinctive chromosome regions is depicted in Appendix A.

Commercially available DNA FISH probes targeting the three chromosomal regions for our Triple DNA FISH assay were selected as defined by aCGH and then used to study touch preparations obtained from frozen tissue of the training set consisting of an equal number (*n* = 33) of cFA and cFTC. Examples of such DNA FISH analyses are shown in Figure 5A–C. One 22q13.3 loss was identified by the Triple DNA FISH assay in a cFA case and not observed by aCGH. This was probably due to some level of tumor heterogeneity, that is, the tiny fraction of tumor cells carrying a particular genetic alteration was easily identified by DNA FISH but below the aCGH detection technical threshold. On the other hand, one case with X gain observed by aCGH was not detected by the Triple DNA FISH assay owing to the poor cellularity of the corresponding touch preparation.

### 3.3. Validation Set on Touch Preparations from Frozen Tissue

Given these encouraging results, we then used the Triple DNA FISH assay to explore the molecular alterations in an independent validation set of 60 samples (60 pts, Figure 1). None of the adenomas carried any of the three tested abnormalities, whereas a total of 19 alterations (three cases of 1p36.31 loss; 12 cases of 22q13.3 loss; and four cases of X gain) were found in 16 pts (47%) with cFTC. In three cases, two alterations were found (1p36.31 loss systematically associated with 22q13.3 loss), and in contrast with the results of the training set, X gain was always found as the only abnormality in our validation set.

### 3.4. Feasibility Set on Archived Already Stained Cytological Samples

The Triple DNA FISH assay was also successfully carried out with the three commercially available probes in a series of 27 archived cytological specimens classified preoperatively as FN/SFN and representing the feasibility set (Appendix A). Six (43%) out of 14 cFTCs but none of the 13 cFAs were found to carry at least one of the three genetic alterations. Two cases corresponding to cFTCs showed loss of the 1p36.31 region, two had 22q13.3 deletions, two samples had X gain, and none had associated double or triple aberrations (Figure 6A,B).

### 3.5. FISH Accuracy

Accuracy characteristics of the Triple DNA FISH assay are detailed in Appendix A in the three individual sets (training: Appendix A, validation: Appendix A, and feasibility: Appendix A), and when they were associated (training plus validation: Appendix A, and training plus validation plus feasibility: Appendix A). As a whole (Appendix A), sensitivity and specificity were 56% (95% CI: 43–67) and 98% (95% CI: 91–100), respectively. Positive predictive values, negative predictive values and accuracy were respectively 98% (95% CI: 85–99), 64% (95% CI: 58–70) and 75% (95% CI: 66–82). Of note, these latter characteristics are dependent on disease prevalence. In our series disease prevalence was 55%. Appendix A shows the same characteristics computed with a disease prevalence of 10%, which may be closer to the real clinical setting.

## 4. Discussion

Our study identifies three distinctive chromosomal imbalances, namely, loss of 1p36.33-1p35.1, loss of 22q13.2-22q13.31 and gain of chromosome X characteristic of half of thyroid cFTCs. Comparison of the aCGH and FISH results showed that when tissue was available for both techniques, alterations detected by aCGH were always confirmed by FISH (Table 2). We also show that these abnormalities are easily detectable by a Triple DNA FISH diagnostic assay on preoperative cytological specimens morphologically classified as FN/SFN.

Among these three either isolated or associated chromosome imbalances, the loss of 22q13.3 was the most frequent (30.3%), followed by the gain of chromosome X (27.27%) and the loss of 1p36.31 (18.18%) (Appendix A). The observation of loss of 22q13.3 is in agreement with reports showing the frequent loss of chromosome 22 in cFTC [47,48,49]. Roque et al. [50] and Liu et al. [51] have suggested that a loss of 22q could represent a potential marker of “aggressive” adenomas. Loss of heterozygosity (LOH) and deletion at chromosome 22 have been reported in FVPTC [51,52,53], an entity we deliberately did not include in our study. The whole chromosome X gain observed in both men and women in our series, which has been reported in one study [47], may suggest that it plays a significant role in cFTC. Interestingly, our study also shows that tumors with X gain had highly aberrant genomes according to aCGH data in the training set (Figure 2A), even though DNA ploidy indices were 1.0 in five out of the six cases (the only DNA aneuploid case had a DNA index of 1.15). In the validation set where no aCGH analysis was performed, the four tumors with X gain detected by DNA FISH were DNA diploid tumors in two cases and DNA multiploid tumors in two cases with DIs of 0.6 and 1.2 and 0.7 and 1.35. The loss of 1p36.31 is in keeping with data reported first by Hemmer et al. [48,49] in 20% of cFTCs and later by Roque et al. [50]. However, the minimal common region in Hemmer’s studies was 1p22-21, in contrast to 1p36.33-1p35.1 in our series.

The global heatmap displayed in Figure 2A show significant differences and also some similarities between cFA and cFTC. It illustrates well the complexity [54] and diversity [55,56,57] of both entities, as well as the concept of genetic instability and of tumor genetic heterogeneity [58,59,60] in which the role of DNA ploidy is a matter of intense debate [61,62]. Differences are striking since the large majority (88%) of cFAs did not exhibit somatic sCNVs, while four (12%) out of 33 did show complex abnormalities (Figure 2A), thereby making their possible premalignant potential questionable. By contrast, most cFTCs showed sCNVs, while 14 (42%) out of 33 did not (Figure 2A), a result that obviously deserves further investigation. Genetic instability is reflected in our DNA flow cytometric and aCGH data: 22 (42%) out of 52 cFTCs and 22 (37%) out of 59 cFAs exhibit DNA aneuploid cell fractions and chromosomal imbalances. Both are detectable in cFTCs and cFAs and more frequent (58%) in the former (19 of 33) than in the latter (12%: 4 of 33) as shown in Figure 2A. These data are in keeping with previous reports [22,47,48,50,51,63,64,65,66,67] and with earlier studies using cytogenetics [68] and conventional CGH [22,47,48,50,51,67] showing that imbalances mainly correspond to whole chromosome gains of chromosome 7 and losses of chromosomes 8, 11, 17 and 18 [50,64,69], respectively; however, losses of chromosome arms more frequently concerned 3p, 11q and 13q, with the loss of 22q being mainly observed in widely invasive cFTCs [48,49]. Similarly, the deletions of the regions harboring known tumor suppressor genes such as *VHL* (3p25-26), *TP53* (17p13) and *PTEN* (10q23) [69,70,71,72,73] and copy number gains of *PIK3CA* on 3q26.32 [74] are mainly affected. Similarities concerning genetic endoreplication were noted in nine (15%) out of 59 cFAs and in nine (17%) out of 52 cFTCs, and DNA haploidy was found only in three (6%) out of 52 cFTCs. Widespread LOH arising from haploidization [30] and of near-haploid chromosomal content has been reported in a large fraction of HCCs, including their metastases [29]. In addition to its widespread physiological role in development and tissue homeostasis across a large range of organisms, endoreplication and polyploidization are known to occur in various neoplasms [75]. Their relevance to genome instability and reversion from polyploidy to diploidy or aneuploidy have been described and may explain some aspects of tumorigenesis through dysregulation of various pathways [76], including EGFR/RAS/MAPK signaling [77]. Of note, it has been recently shown that whole-genome doubling buffers the impact of deleterious alterations [78] and that single-chromosomal gains can function as promoters and metastasis suppressors [79].

Differences and similarities between cFTC with cPTC and FVPTC have also been described. Differences concern patterns of upregulation and downregulation of gene [80,81,82,83,84,85], mRNA and miRNA [20,86,87,88] expression and chromosomal aberrations. First, cPTC and FVPTC have a low prevalence of chromosomal aberrations and the majority of tumors show no evidence of genetic instability [22,52,89]. Second, cPTC has a relatively quiet genome overall, as shown by conventional cytogenetics [64,90,91], frequent (approximately 80%) DNA diploidy detected by flow cytometry [65,66], a low rate of LOH [63] and chromosomal imbalances detected in 40% of cases by CGH [47,48,49,50,92,93,94,95,96], aCGH [67,89,97], SNP array [51] or whole-exome sequencing [98]. Third, cPTC has one of the lowest overall somatic mutational burdens among all cancer types [99,100]. Similarities, however, are noticed when comparing our data and those derived from the large-scale next-generation approach performed by the TCGA consortium [20]. For example and in addition to the absence of chromosome arm-level alterations in 73% of cPTCs, the remaining tumors were also classified into three groups: a first one had single-arm imbalances consisting of cases (14%) with an isolated 1p gain and a few other sCNVs enriched for PTC tall cell variants and *BRAF*^V600E^ mutations; a second one (10%) had an isolated 22q loss enrichment in FVPTC; and a third one comprised few cases (3%) with a complex pattern of losses and gains involving multiple chromosome arms [20], in line with our set of cFTCs (39%) exhibiting complex sCNVs (Figure 2A). Of note, the tendency of cFTCs to have no driver mutations or fusions parallels the existence of sCNVs, which may in turn suggest that sCNV by itself represent a driver event initiating the development of a subset of cFTCs.

We could not confirm loss of chromosome 9 in cFTC that contrasted with its gain in cFA [47] and did not find evidence for alterations of *PKCε* on 2p21, a candidate gene thought to be involved in thyroid tumorigenesis [101]. Furthermore, our findings are at variance with an aCGH study showing that: monosomy 21 appears to differentiate 100% of adenomas from carcinomas; trisomy 14 differentiates 100% of minimally invasive carcinomas from adenomas; and trisomy 10 identifies 100% of minimally invasive carcinomas from widely invasive carcinomas [102].

Our feasibility study further substantiated the application of the Triple DNA FISH assay on archived cytological material. For this purpose, we selected cytological slides with storage times varying between 1 month and 12 years, which opens the possibility for large retrospective studies. All 27 cases could be concluded, and the results found are in excellent agreement with our previous findings based on our training and validation sets. An evaluation algorithm depicting the main data interpretation steps of the Triple DNA FISH assay is shown in Figure 7. This algorithm could be applied in the cytomorphological differential diagnosis of FN/SFN sampled by FNAC. We contend that a panel including 1p36.31, 22q13.3 and centromere X with 1q25.3, 22q11.2 and centromere Y as controls represents a potential useful first-line adjunct of certain follicular tumors on initial cytological evaluation. Particular emphasis is advised in male patients with X gain who show two X signals and one Y signal in that the FISH result should be considered “trisomic”. When the interpretation of the FISH results is unclear, DNA measurement by flow or static cytometry may be needed. However, in this case, other types of molecular tests could also be performed [103], or the patient could be considered a good candidate for conservative surgery, such as diagnostic lobectomy [1,10].

The cytomorphological challenges encountered in Bethesda categories III and IV have indeed prompted scientists to develop preoperative genomic panels adapted to cytological samples obtained by FNAC, the so-called “rule-in” and “rule-out” tests [104]. The aim is to prevent the patient from repeating FNAC following an AUS/FLUS diagnosis or performing a diagnostic lobectomy with isthmusectomy in case of a diagnosis of FN/SFN [1,10] by optimizing surgical management and reducing the avoidable treatment of benign nodules [3,104,105,106].

The ThyroSeq^®^ v3 Genomic Classifier (CBL Path, Inc., Rye Brook, NY, USA and University of Pittsburgh Medical Center, Pittsburgh, PA, USA) is a “rule-in” method [107] based on targeted next-generation sequencing of DNA and RNA validated in a multicentric, prospective and blinded study [108]. The Afirma^®^ Genomic Sequencing Classifier (GSC) (Veracyte, Inc., South San Francisco, CA, USA) is a next-generation “rule-out” method based on the previous Gene Expression Classifier validated first in a multicentric, prospective and blinded study [109] then with RNA-seq methodology [110]. The ThyGeNEXT/ThyraMIR^®^ (Interspace Diagnostics, Inc., Parsippany, NJ, USA) is a combined method detecting gene mutations and fusions (ThyGeNEXT^®^) and a complementary miRNA expression classifier (ThyraMIR^®^) that has been clinically validated [111]. The Rosetta GX reveal™ Thyroid Classifier (Rosetta Genomics Philadelphia, PA, USA), no longer commercially available, was a validated method to evaluate the expression pattern of a series of miRNAs directly extracted from stained slides of preoperative FNAC [112]. The cost-effectiveness of the different molecular tests versus lobectomy is a matter of debate [113,114,115], and the main disadvantage of these methods is their cost, which makes them rarely used in Europe [116]. In addition, these tests were first mainly developed to identify cPTC and its variants and then to detect medullary thyroid carcinoma, Hürthle cell neoplasms and parathyroid tissue (Afirma^®^, ThyroSeq^®^, ThyGeNEXT/ThyraMIR^®^) but not to specifically identify cFTC.

The advantages of our customized Triple DNA FISH diagnostic assay are multiple: it is a direct, rapid, and low-cost technique that is easy to perform in a routine laboratory equipped with basic fluorescence microscopy; it may be performed as the first-line approach on preoperatively stained slides with a cytological diagnosis of FN/SFN, obviating the need to perform an additional instant or delayed collection of material by FNAC for molecular testing; and it is a well-standardized diagnostic procedure that is adapted to the identification of cFTC and may be carried out on only a few cells, with the critical advantage of single-cell study.

There are several limitations in our study: its retrospective nature and the rate of sample exclusion in both the training (14%) and validation (12%) sets, which may introduce some bias in the statistical analysis, and the relatively small number of preoperative FNAC specimens (*n* = 27) tested. Although bicentric and showing a very high specificity (close to 100%) at the expense of a low (50%) sensitivity, our work obviously needs further prospective and multi-institutional validation and postvalidation studies in a routine clinical setting; the use of a relatively old oligonucleotide aCGH technology (Agilent 244K Whole Human Genome (G4411B)) is an intrinsic limit: the rapid evolution of current high-throughput technologies such as NGS may obviously and easily overcome this limit. However, given the size of the two region losses characterized, trying to more precisely define the subregions on chromosomes 1p and 22q and a gain of the entire chromosome X may not be of actual interest; the presence of one case harboring a *BRAF*^600E^ mutation is puzzling: this mutation is associated with cPTC in about half of the cases and in a higher percentage of the hobnail and tall cell variants, may be present in some FVPTCs, but should not be present in cFTCs, suggesting the possible misclassification of FVPTC (which was not observed after reviewing the corresponding sections again); the restrictive analysis of FN/SFN (Bethesda category IV) within the spectrum of indeterminate cases (gray zone) is an additional limitation of our study: according to TBSRTC [11,12], indeterminate cases comprise two other categories: AUS/FLUS (Bethesda category III) and SM (Bethesda category V). However, we deliberately focused on the FN/SFN (Bethesda IV) category because its pattern is well defined [117]; the AUS/FLUS category is still subject to significant interobserver variability [118,119,120,121,122,123,124] and may contain a significant number of FVPTCs [125]; and the SM category mainly corresponds to cPTC or variants of PTC [12].

In summary, we identified three distinctive chromosomal imbalances in a set of 153 pts with thyroid follicular lesions using aCGH and developed a Triple DNA FISH diagnostic assay. An evaluation algorithm to better distinguish between thyroid cFA and cFTC using our assay is presented in Figure 7. This diagnostic assay detects structural abnormalities that were observed in approximately half of the cFTCs. It is easy to perform on archived cytological materials and its specificity higher than 98% might be comparable with *BRAF*^600E^ testing in cases of suspicion of PTC. In the context of challenges encountered in both ultrasonographic analyses [126,127,128,129,130,131,132,133,134,135] and indeterminate cytomorphology [118,119,120,121,122,123,124] as well as global financial restrictions, we believe our diagnostic assay merits further attention.

## 5. Conclusions

These data suggest that our triple DNA FISH diagnostic assay may detect 50% of cFTCs with a specificity higher than 98% and be useful as a low-cost adjunct to cytomorphology to help further classify follicular neoplasms on already routinely stained cytological specimens.

## Figures and Tables

**Figure 1 cancers-12-02529-f001:**
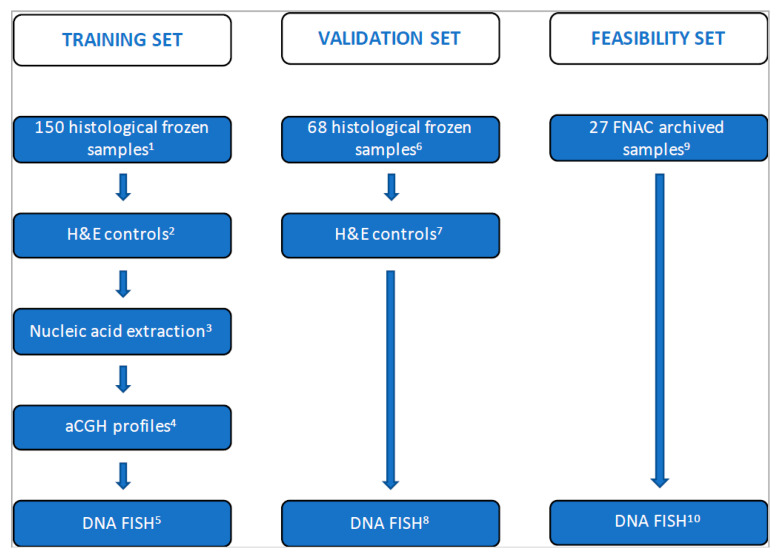
Flow chart of the study. Abbreviations: pts: patients; H&E: hematoxylin & eosin; aCGH: array comparative genomic hybridization; DNA FISH: DNA fluorescent in situ hybridization; hyb: hybridization; FNAC: fine needle aspiration cytology. Training set: ^1^ from 87 pts; ^2^ from 69 pts (after exclusion of 18 cases: 10 atypical adenomas, four oxyphilic neoplasms, two uncertain diagnoses, one follicular variant of papillary carcinoma, one non-tumoral); ^3^ from 67 pts (after exclusion of two cases with insufficient DNA for analysis); ^4^ from 66 pts (after exclusion of one case with doubtful aCGH result); ^5^ from 44 pts (with available residual tissue). Validation set: ^6^ from 68 pts; ^7^ from 60 pts (after exclusion of eight cases: six oxyphilic neoplasms, one atypical adenoma, one metastatic tumor); ^8^ from 60 pts. Feasibility set: ^9^ from 27 pts (preoperative routinely stained FNAC with histological assessment of the corresponding postoperative tissue); ^10^ from 27 pts.

**Figure 2 cancers-12-02529-f002:**
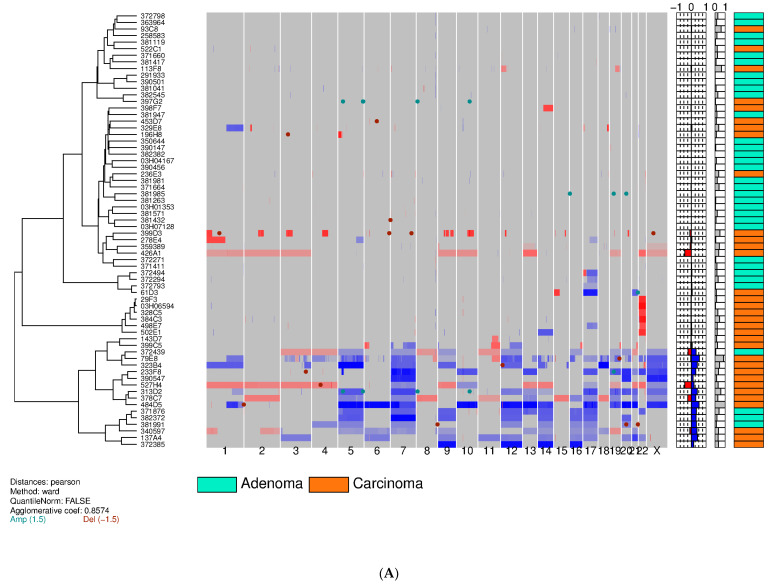
(**A**) Global heatmap of 66 clustered samples (training set: 33 cFTCs and 33 cFAs) using the Pearson distance and Ward’s method. Each row represents a patient, and each column represents a chromosome. (**B**) Frequency plots of gains and losses for each chromosomal region of 66 samples. The red- and blue-colored bars correspond to the percentage of samples with losses or gains, respectively, at each indicated region. Abbreviations: Adenoma: classic follicular adenoma (cFA); Carcinoma: classic follicular thyroid carcinoma (cFTC).

**Figure 3 cancers-12-02529-f003:**
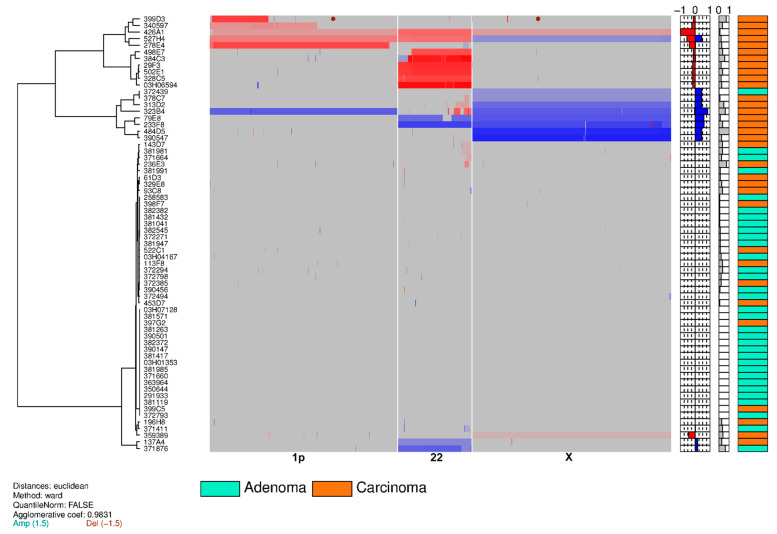
Heatmap of the three chromosomes/chromosome arms that can be used to distinguish between cFA and cFTC tissues in the training set of 66 samples: chromosomes 1p, 22 and X (Euclidean distance, Ward’s method). Abbreviations: Adenoma: classic follicular adenoma (cFA); Carcinoma: classic follicular thyroid carcinoma (cFTC).

**Figure 4 cancers-12-02529-f004:**
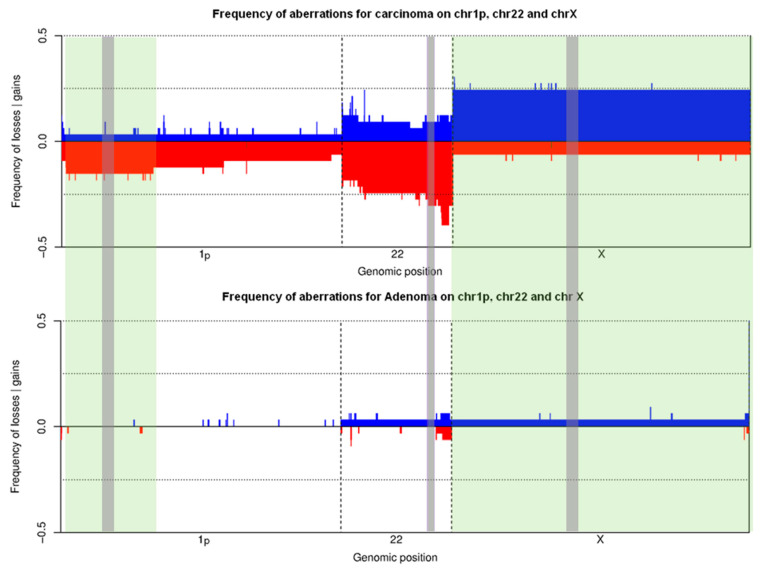
Localization of the minimum common regions of the chromosomal aberrations in cFA and cFTC as detected by aCGH (vertical green bands: loss of 1p36.33-1p35.1 and 22q13.2-22q13.31 and gain of X) and by commercially available DNA FISH probes specific to the 1p36.31, 22q13.3 loci and to the centromere of the X chromosome (vertical gray bands). Abbreviations: aCGH: array comparative genomic hybridization; Adenoma: classic follicular adenoma (cFA); Carcinoma: classic follicular thyroid carcinoma (cFTC); DNA FISH: DNA fluorescent in situ hybridization.

**Figure 5 cancers-12-02529-f005:**
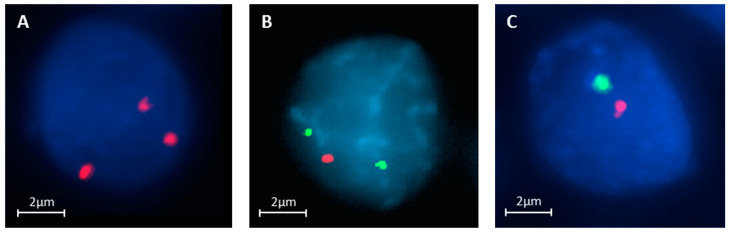
The Triple DNA FISH diagnostic assay illustrating the most common characteristic alterations detected in cFTC. (**A**) The three red signals show an X gain (female patient, Vysis CEP X/Y Abbott Laboratories, Abbott Park, IL, USA). (**B**) The two control green signals (1q25.3) and one red signal that is specific for 1p36.31 correspond to a loss of the 1p region (ZytoLight 1p36/1q25, ZytoVision, Bremerhaven, Germany). (**C**) One green signal (22q13.3) and one red signal (22q11.2) show a loss at 22q (LSI TUPLE 1/LSI ARSA, Abbott Laboratories). Abbreviation: cFTC: classic follicular thyroid carcinoma.

**Figure 6 cancers-12-02529-f006:**
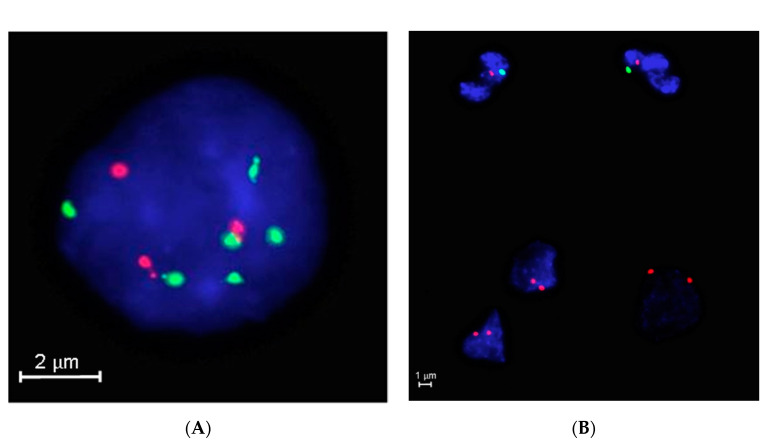
The Triple DNA FISH diagnostic assay on archived cytological samples of cFTC. (**A**) The six control green signals (1q25.3) and three red signals that are specific for 1p36.31 correspond to a loss of the 1p region (ZytoLight 1p36/1q25, ZytoVision, Bremerhaven, Germany). (**B**) The two red signals of the tumor cells show an X (red) gain and Y (green) loss next to normal granulocytes showing one red (X) and one green (Y) signal (male patient, Vysis CEP X/Y Abbott Laboratories, Abbott Park, IL, USA). Abbreviation: cFTC: classic follicular thyroid carcinoma.

**Figure 7 cancers-12-02529-f007:**
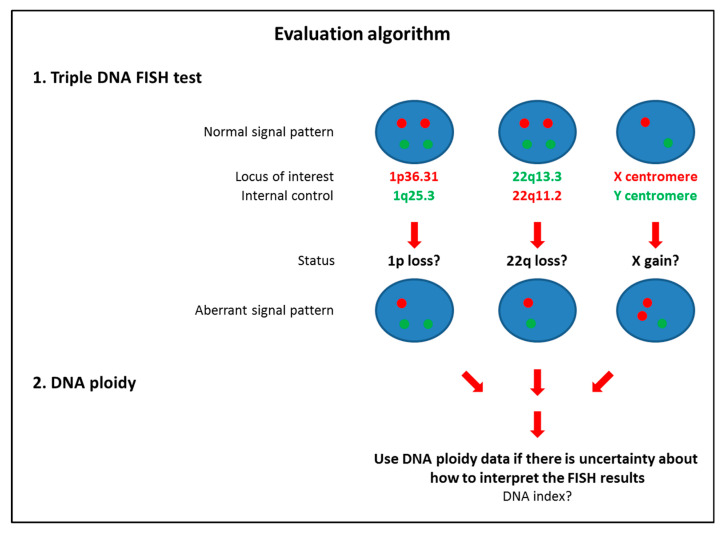
The evaluation algorithm of the Triple DNA FISH diagnostic assay depicts the main three normal and aberrant signal patterns observed in samples from cFA and cFTC, respectively. Abbreviations: cFA: classic follicular adenoma; cFTC: classic follicular thyroid carcinoma; DNA FISH: DNA fluorescent in situ hybridization.

**Table 1 cancers-12-02529-t001:** Clinicopathological and biological characteristics of the 126 patients with follicular thyroid neoplasms (training and validation sets).

Characteristics	Unit	Training Set			Validation Set			Total		
		Adenoma	Carcinoma	Total	Adenoma	Carcinoma	Total	Adenoma	Carcinoma	Total
Patients	Number	33 (50%)	33 (50%)	66 (100%)	26 (43%)	34 (57%)	60 (100%)	59 (47%)	67 (53%)	126 (100%)
Gender:	Number									
Female		21 (64%)	16 (48%)	37 (56%)	20 (77%)	20 (59%)	40 (67%)	41 (69%)	36 (54%)	77 (61%)
Male		12 (36%)	17 (52%)	29 (44%)	6 (23%)	14 (41%)	20 (33%)	18 (31%)	31 (46%)	49 (39%)
Age med. (range)	Years	52 (21–70)	53 (25–78)	52 (21–78)	47 (27–75)	54 (21–82)	51 (21–82)	50 (21–75)	53 (21–82)	52 (21–82)
T size med. (range)	cm	3 (1–6.4)	4.5 (1–15)	3.8 (1–15)	2.8 (1.5–9)	4.4 (1.5–11)	3.7 (1.5–11)	2.9 (1–9)	4.5 (1–15)	3.7 (1–15)
Follow-up med. (range)	Months	35 (3–180)	47 (0–140)	41 (0–180)	12 (1–48)	42 (2–120)	29 (1–120)	25 (1–180)	44 (0–140)	35 (0–180)
Clinical status	Number, alive	32 (100%)	30 (91%)	62 (95%)	26 (100%)	30 (88%)	56 (93%)	58 (100%)	60 (90%)	118 (94%)
Mutations:										
*KRAS/*NA		0 (0%)/1	0 (0%)/1	–	1 (5%)/4	0 (0%)/5	–	1 (2%)/5	0 (0%)/6	–
*HRAS/*NA		1 (3%)/1	5 (16%)/2	–	1 (4%)/2	6 (20%)/4	–	2 (4%)/3	11 (18%)/6	–
*NRAS/*NA		2 (6%)/1	6 (20%)/3	–	4 (17%)/2	8 (27%)/4	–	6 (11%)/3	14 (23%)/7	–
*BRAF/*NA		0 (0%)/1	1 (3%)/1	–	0 (0%)/2	0 (0%)/4	–	0 (0%)/3	1 (2%)/6	–
*CTNNB1/*NA		0 (0%)/1	0 (0%)/1	–	–	–	–	0 (0%)/1	0 (0%)/1	–
*PIK3CA/*NA		0 (0%)/1	0 (0%)/3	–	–	–	–	0 (0%)/1	0 (0%)/3	–
Translocations:										
*PAX8/PPARγ1/*NA		0 (0%)/1	3 (12%)/7	–	0 (0%)/2	2 (6%)/1	–	0 (0%)/3	5 (8%)/8	–
*RET/PTC1/*NA		0 (0%)/0	0 (0%)/7	–	0 (0%)/0	0(0%)/0	–	0 (0%)/0	0 (0%)/7	–

NA: not available.

**Table 2 cancers-12-02529-t002:** Detection of the three genetic anomalies in classic follicular adenoma (cFA) and carcinoma (cFTC) in the training, validation and feasibility sets: comparison between aCGH and Triple DNA FISH assay.

Chromosome	Training Set		Validation Set		Feasibility Set		Total	
	Adenoma aCGH (%)/FISH (%)	Carcinoma aCGH (%)/FISH (%)	Adenoma aCGH (%)/FISH (%)	Carcinoma aCGH (%)/FISH (%)	Adenoma aCGH (%)/FISH (%)	Carcinoma aCGH (%)/FISH (%)	Adenoma aCGH (%)/FISH (%)	Carcinoma aCGH (%)/FISH (%)
1p36.31 loss	0 (0%)/0 (0%)	5 (15%)/5 (15%)	–/0 (0%)	–/3 (9%)#	–/0 (0%)	–/2 (14%)	0 (0%)/0 (0%)	5 (6%)/10 (12%)
22q13.3 loss	0 (0%)/1 (3%)	8 (24%)/8 (24%)	–/0 (0%)	–/12 (35%)#	–/0 (0%)	–/2 (14%)	0 (0%)/1 (1.4%)	8 (9.9%)/22 (27%)
X gain	1 (3%)/0 (0%) *	8 (24%)/8 (24%) **	–/0 (0%)	–/4 (12%)#	–/0 (0%)	–/2 (14%)	1 (1.4%)/0 (0%) *	8 (9.9%)/14 (17.3%) **
Total	2 alterations (1 aCGH, 1 FISH) in 2 patients (6%)	22 alterations (21 aCGH, 21 FISH) ** in 18 patients *** (55%)	0 alteration (FISH) (0%)	19 alterations (FISH) in 16 patients (47%)	0 alteration (FISH) (0%)	6 alterations (FISH) in 6 patients (43%)	2 alterations (1 aCGH, 1 FISH) in 2 patients (2.8%)	47 alterations (21 aCGH, 46 FISH) ** in 40 patients (49.4%)

Abbreviations: aCGH: array comparative genomic hybridization; FISH: fluorescent in situ hybridization. * The case was positive for X gain by aCGH but was not available for FISH. ** Out of the eight cases that were positive for X gain by FISH, seven cases also showed X gain by aCGH, while one X-gain positive by aCGH was not available for FISH and one X-gain positive by FISH was not detected by aCGH. *** One sample was not available for FISH. The sample that was positive for X gain only using FISH also carried a 1p loss that was also detected with aCGH and FISH.

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
