# Peer review of "DNA FISH Diagnostic Assay on Cytological Samples of Thyroid Follicular Neoplasms†"

_cancers, 2020, doi:10.3390/cancers12092529_

Round 1

Reviewer 1 Report

This manuscript is well written. The materials and methods, approach, statistical analysis are all sound.  As well as addressing the limitations.  However, not sure if there are a lot of interests to the audience as there are currently many studies focusing on molecular profiling  to distinguish between FA and FTC. 

Author Response

Thank you for the comments.

We think that our study may be very interesting for pathologists dealing with thyroid cytopathology. 

Reviewer 2 Report

This is an interesting, thorough and well-written article on DNA FISH diagnostic Assay on cytological samples of thyroid follicular neoplasm. The authors show the feasibility of the DNA FISH assay in samples obtained using fine-needle aspiration cytology (FNAC). The authors suggest that their triple DNA FISH diagnostic assay may detect 50% of follicular thyroid carcinomas (FTCs) and be useful as a low-cost adjunct to cytomorphology to help further classify follicular neoplasms on already routinely staine FNAC samples. All this is correct, but some data makes the study less robust. The article has two main weaknesses: the relatively small number of samples used and the incomplete histopathological evaluation of the samples.

As the authors describe very well in the introduction, if one excludes BRAFV600E in papillary carcinoma (PTC) with papillary pattern, there are no molecular alterations that can confidently differentiate between follicular adenoma, PTC follicular variant and / or FTC. Although these follicular pattern tumors share the same type of molecular alterations, these alterations appear in a higher percentage in malignant tumors. One wonders if the same would happen in this study using a larger number of samples. In fact in this work (Table 2), 2 patients with follicular adenoma showed genetic abnormalities (1 aCGH [X gain] and 1 FISH [22q13.3 loss]). Authors should comment on this possibility explicitly in the discussion.

At present, the histopathological study is the gold standard for the diagnosis of thyroid lesions. Histopathological diagnosis is a key reference to establish the further treatment of patients with thyroid tumors, the prognosis and the guide for clinical trials. In this article, the little attention paid to the histopathological data is striking in comparison with the thoroughness devoted to the other aspects of the study. This is a very important weakness of the study. According to the 2017 WHO classification criteria, the authors eliminated Hürthle cell neoplasms from the study, but did not adopt this classification to classify FTC cases (minimally invasive, encapsulated angioinvasive and widely invasive FTCs). This fact is a relevant limitation since FTCs showing only a capsular invasion (minimally invasive FTC) have an excellent prognosis and can be treated more conservatively. The article should show a table (and comments) of the global results in relation to at least 2 histopathological subgroups of FTC: I) of low aggressiveness (FTCs showing only capsular invasion) and II) the other more aggressive group (FTCs with vascular invasion and / or widely invasive).

The authors describe the presence of a case that harbors a BRAFV600E mutation, for which they themselves suggest a possible misclassification of the PTC follicular variant, subsequently, the same authors discarded PTC after reviewing the sections again. One wonders if the revision of this particular case (AF or FC?) has been carried out on frozen or paraffin sections. In frozen tissue sections, recognition of the nuclear characteristics of PTC may be difficult or inconclusive. Was this case associated with venous invasion or lymph node metastasis?. Authors should specify whether all cases included in the study (paraffin sections) have been reviewed by at least one pathologist.

In Table 1, the clinical status should include the number of deaths directly related to the FTC.

Author Response

Thanks for the detailed comments of the Reviewer.

Concerning the first point, we would like to emphasize that, to the best of our knowledge, our study of 153 cases is the largest series ever published yet. Therefore the comment may be right but should be based on facts. The presence of molecular and/or genetic abnormalities in cFTA is well known and the findings of 2 molecular abnormalities has been clearly stated in the Discussion (page 16, lines: 78-80): "Differences are striking since the large majority (88%) of cFAs did not exhibit somatic CNVs, while 4 (12%) of 33 did show complex abnormalities (Figure 2A), thereby making their possible premalignant potential questionable."

We fully agree that histopathology is the current gold standard for definitive diagnosis, clinical stratification and for guiding therapy. However, the comment about "the little attention paid to the histopathological data is striking..." as well as the statement about the absence of reclassification of tumors according to the 2007 WHO classification is surprising since we have stated in the Materials and Methods section (page 3, lines: 126-129): "The initial diagnosis of cFTC performed on paraffin-embedded histopathological samples [37] was kept and not reclassified according to the most recent 2017 WHO classification of tumors of endocrine organs [16] as well as the substratification according to the number of veins (<4 vs. ≥4) invaded [38]." All tissue samples included in our series were indeed obtained before 2007 and our decision of not reclassifying them according to WHO 2007 and not including Hürthle cell tumors and other neoplasms such as FVPTC for example, although arguable, was deliberate and included in the design of the study. The distribution of cFA and cFTC subtypes is not shown in Table 1, but mentioned in the Results paragraph 3.1.: "Histopathological subtypes [16] of cFA (n = 59) showed a microfollicular, macrofollicular, and mixed architectural pattern of growth in 19, 13 and 27 pts, respectively (without fetal/solid type), whereas cFTCs (n = 67) with various growth patterns were minimally invasive (capsular invasion only) and widely invasive in 24, 10 and 33 pts, respectively (data not shown). We would like to stress that our purpose was not to develop a tool for prognostic purposes, to help distinguish minimally and widely invasive cFTC or to guide therapy, but to set up a diagnostic assay, applicable to preoperative cytological samples classified as Bethesda IV, and capable triaging patients who are the best candidates for surgery. 

The revision of the case of cFTC with a BRAFV600E mutation has been done on paraffin-embedded tissue and that case was not associated with lymph node metastasis. As mentioned in the Materials and Methods, histopathology was done by 4 senior pathologists (AEN, BC, AAG, PV) in major comprehensive cancer centers (MDACC, Houston, TX, USA and Gustave Roussy, France).

Because we were not always able to verify that death of patients were strictly related to cFTC the clinical status of the patients mentioned in Table 1 only mention the number of patients alive at the time of the study. 

In the introduction, the reviewer mentions two main weaknesses in our study i.e. the relatively small number of samples used and the incomplete histopathological evaluation of the samples. We do not agree with the former and think we have explained the reasons of the latter. 

Reviewer 3 Report

The study presents the building up and validation of a FISH diagnostic assay for lesions classified cytologically as follicular neoplasms (Bethesda IV) whose histological counterpart should be mostly represented by follicular adenoma (FA), follicular thyroid carcinoma (FTC) and follicular variant of papillary thyroid carcinoma (FVPTC). The latter was not considered in the study as well as oncocytic lesions.

The topic is relevant because cytologically indeterminate lesions are still a major problem in the context of thyroid cancer. Although diagnostic tests useful for driving surgical decisions of these lesions already exist, they are quite expensive.

Overall, the design of the study is clear, experimental procedures seems well done and the results are interesting and well presented.

However, there are some aspects that could be improved:

  • The presence of a BRAF V600E mutation in FTC is quite unusual. The authors declare that the original histological diagnosis was kept, and lesions were not reclassified according to 2017 WHO criteria. I think that histological glass slides should be carefully revised and maybe re-classified. Since it is very likely that the lesion harbouring the BRAF V600E mutations will turn out to be a FVPTC, it should be removed since this was a criterion of selection of the authors. In addition, I wonder if FA presenting gross aberrations do not present any sign of invasion; were they sampled in toto?
  • The reporting of the number of MI-FTC and WI-FTC (first line of page 8) is confusing. It seems that there are two groups (but three numbers are reported), and minimally invasive are tumors with capsular invasion only. I think I understand what the authors would have meant, but it is not clear. In addition, the histologic criteria considered should be reported in the methods.
  • Are the samples of the feasibility set paired to those used in the training and/or validation sets? It should be specified.
  • Please, report the method used for identifying the minimal common regions differentiating FTC from FA.
  • Sensitivity and specificity should be reported for all the three sets. Although these are selected series, PPV and NPV should be reported. Moreover, the P-value for sensitivity and specificity is not very informative; on the contrary, confidence intervals should be reported.
  • Table 2 should include also the results of the validation and the feasibility sets.
  • It would be interesting to perform a direct comparison of the diagnostic performance between the FISH assay and the mutations detected.
  • Since MI-FTC and WI-FTC have a very different prognosis, it would be interesting to compare them in terms of genomic aberrations. Moreover, since the authors have the follow-up data, were these alterations associated with prognosis?
  • The authors stated that one glass slide per case was used for the FISH assay. Since all three sets of probes are in the green/red fluorescence channel, was the glass split into three fields? Please add some details to help the reader understand.
  • The authors conclude that the FISH assay “is the equivalent, in terms of sensitivity and specificity, of BRAF V600E testing in cases of suspicion of PTC”. Although the specificity of the assay herein reported is high, BRAF V600E mutation (with few exceptions in the Asian population) has a specificity very close to 100%. Moreover, the FISH assay was tested on a limited number of cases; then, I suggest that the sentence should be softened.
  • DICER1 mutations can be found also in FVPTC both in sporadic tumors (one of the three DICER1-mutant cases of TCGA was a FVPTC) and in the context of a DICER1 Please modify this topic in the introduction.
  • There are a few slips: in the introduction, the acronym FLUS was spelled as “follicular atypia of undetermined significance” instead of follicular lesion; in the introduction, the RET/PTC3 rearrangment was called PET3; in the discussion, the TCGA consortium is reported as “TCGH”.

Author Response

Thanks to the Reviewer for the detailed comments.

We agree that the presence of a BRAF V600E mutation in FTC is quite unusual. It is suggested to review the slides and "may be" reclassified the case. As mentioned in the paragraph of limitations (Discussion section) : "the presence of one case harboring a BRAF600E mutation is puzzling. This mutation is associated with cPTC in about half of the cases and in a higher percentage of the hobnail and tall cell variants, may be present in some FVPTCs but should not be present in cFTCs, suggesting the possible misclassification of FVPTC (which was not observed after reviewing the corresponding sections again)". Since histopathology was diagnosed and reviewed by all four senior histopathologists from two major comprehensive cancer centers co-authoring the manuscript, the possibility of misclassification although possible should be quite low and we don't think this case should be removed based only on an unexpected molecular event. Concerning cFA all of them were extensively sampled according to the recommended guidelines and none of them showed sign of invasion.

We agree that the numbers of minimally and widely invasive cFTCs are confusing (3 numbers for 2 categories). The number 10 has been deleted. In addition, as requested, histologic criteria are reported and highlighted in yellow (M&M section, paragraph 2.1.).

The samples of the feasibility set were paired to those used in the training and/or validation sets: this is now specified and highlighted in yellow (M&M section: 2.1.) as well the method used for identifying the minimal common regions differentiating FTC from FA (M&M section, paragraph 2.4.).

As requested, sensitivity, specificity, PPV and NPV with their respective confidence intervals are now reported in Supplementay Table 3 and highlighted in yellow within the manuscript (Results section, paragraphs 3.4. and 3.5. modified). Since our series does not reflect the real clinical setting (disease prevalence of cFTC around 50%), Supplementary Table 3 also includes the same characteristics computed with a disease prevalence of 10%. A sentence has been added at the end of paragraph 3.5. (Results section) and the old text mentioning P-values has been deleted.

Concerning the anomalies in the training set and according to the recommendation of the Reviewer, the old Table 2 now highlighted in green in the manuscript may be replaced by the new submitted Table 2.

It is suggested to perform a direct comparison of the diagnostic performance between the FISH assay and the mutations detected. However, there are no mutations specific for cFTC. Comparison with the anomalies detected by FISH that are very suggestive of cFTC seems therefore too artificial in our opinion.

According to the Reviewer's comment, we looked at the genomic aberrations of minimally and widely invasive FTC and did not find major differences. More precisely, of 17 minimally invasive FTC analyzed by aCGH, anomalies were: absent (4 cases), minimal (6 cases, including 2 with loss of 22q), numerous (7 cases, including loss of 1p and 22q and gain of X), and; of 15 widely invasive FTC analyzed by aCGH, anomalies were: absent (2 cases), minimal (6 cases, including 3 losses of 22q), intermediate (2 cases) and numerous (5 cases, including 4 with X gain). This is now mentioned in the manuscript and highlighted in yellow (Results section, paragraph 3.2.). We did not compare genetic alterations with follow-up data as we only know the patients' status (alive vs dead) but are not sure about its strict relationship with cFTC outcome. 

To answer the question about the three sets of probes, we used one glass slide to perform the triple DNA FISH test by splitting one glass into three different fields for the 3 in situ hybridizations performed under 3 different small round coverslips. This is now added and highlighted in yellow in the manuscript (M&M section, paragraph 2.5.).

In our conclusion, the sentence about the FISH assay has been softened as requested by the Reviewer (highlighted in yellow, Discussion section).

We agree with the Reviewer that in the introduction the DICER1 mutations can be found also in FVPTC both in sporadic tumors (one of the three DICER1-mutant cases of TCGA was a FVPTC) and in the context of a DICER1. We have modified the sentence accordingly by replacing "all except for eFVPTC and iFVPTC may carry DICER1 alterations [25]" by "all except for eFVPTC may carry DICER1 alterations" (highlighted in yellow in the Introduction section).

Thanks to the Reviewer for mentioning the other slips: follicular lesion of atypical significance and RET/PTC3 rearrangement (both in the Introduction section), and TCGA (Discussion section): they are modified and highlighted in yellow accordingly.

As there is no possibility to upload more than one file, the final manuscript is enclosed. I send by e-mail the required new Table 2 and a Supplementary table 3 to the MDPI Assistant Editor, Ms. Cosmina Vircan.
